# Maternal and paternal employment in agriculture and early childhood development: A cross-sectional analysis of Demographic and Health Survey data

Lilia Bliznashka [1,2]*, Joshua Jeong [3], Lindsay M. Jaacks [2]

1 International Food Policy Research Institute, Washington, DC, United States of America, 2 Global Academy of Agriculture and Food Systems, University of Edinburgh, Midlothian, United Kingdom, 3 Department of Global Health and Population, Harvard T.H. Chan School of Public Health, Boston, MA, United States of America

* l.bliznashka@cgiar.org

**Data Availability Statement:** The data underlying the findings presented are publicly available from the DHS Program (http://www.dhsprogram.com).

## Abstract

Considerable literature from low- and lower-middle-income countries (LLMICs) links maternal employment to child nutritional status. However, less is known about the role of parental employment and occupation type in shaping child development outcomes. Additionally, little empirical work has examined the mechanisms through which parental occupation influences child outcomes. Our objective was to investigate the associations between maternal and paternal employment (comparing agricultural and non-agricultural employment) and child development and to examine childcare practices and women's empowerment as potential mechanisms. We pooled nine Demographic and Health Surveys (Benin, Burundi, Cambodia, Congo, Haiti, Rwanda, Senegal, Togo, and Uganda) with data on 8,516 children aged 36–59 months. We used generalised linear models to estimate associations between parental employment and child development, child stimulation (number of activities provided by the mother, father, and other household members), child supervision (not left alone or with older child for >1 hour), early childhood care and education programme (ECCE) attendance, and women's empowerment. In our sample, all fathers and 85% of mothers were employed. In 40% of families, both parents were employed in agriculture. After adjusting for child, parental and household confounders, we found that parental agricultural employment, relative to non-agricultural employment, was associated with poorer child development (relative risk (RR) 0.86 (95% CI 0.80, 0.92), more child stimulation provided by other household members (mean difference (MD) 0.26 (95% CI 0.09, 0.42)), less adequate child supervision (RR, 0.83 (95% 0.78, 0.80)), less ECCE attendance (RR 0.46 (95% CI 0.39, 0.54)), and lower women's empowerment (MD -1.01 (95% CI -1.18, -0.84)). Parental agricultural employment may be an important risk factor for early childhood development. More research using more comprehensive exposure and outcome measures is needed to unpack these complex relationships and to inform interventions and policies to support working parents in the agricultural sector with young children.

Surveys used in this paper are provided in the Supporting Information files. Registration is required to access the data.

**Funding:** This research was funded in whole, or in part, by the Medical Research Council/UK Research and Innovation, https://www.ukri.org/councils/mrc/, grant number: MR/T044527/1 to LJ. The funder had no role in study design, data collection and analysis, decision to publish, or preparation of the manuscript.

**Competing interests:** The authors have declared that no competing interests exist.

## 1. Introduction

In low- and lower-middle income countries (LLMICs), up to 40% of children are developmentally off-track [1]. Improving child development in early life can improve adult educational, labour, and health outcomes [2, 3]. Prior literature has examined many biopsychosocial (e.g., nutritional deficiencies, suboptimal childcare practices) and contextual (e.g., societal violence) risk factors for poor child development [4]. However, parental employment, which affects caregivers' capacities and children's ecological environments, remains understudied in relation to child development even though 61% of adults in LLMICs participate in the labour force [5].

### 1.1. Parental employment and child nutritional outcomes

To the best of our knowledge, no studies to date have examined the associations between parental occupation and child development in LLMICs. Instead, numerous studies have examined the association between maternal employment and child nutritional status and yielded mixed results. In some cases, maternal employment is associated with improved child nutritional outcomes [6–8], whereas in other cases maternal employment is associated with poor child nutritional outcomes [9–14]. Still other studies have found no significant associations between maternal employment and child nutritional outcomes [15–17]. Similarly, findings with respect to maternal occupation type are also mixed. In some contexts, maternal agricultural and manual employment are associated with poor child nutritional outcomes [9], whereas in others, maternal non-agricultural employment is associated with poor child nutritional outcomes [18].

Importantly, none of these studies in LLMICs considered the associations between *paternal* employment or occupation type and child outcomes. This is despite prior work highlighting that fathers are often the primary breadwinner of the family, providing substantial financial resources that can support positive outcomes for all household members [19, 20]. Only a few known studies have investigated the role of both maternal and paternal employment in child outcomes [21, 22]. The emerging evidence is largely inconclusive and none of the studies considered child development as an outcome. For example, one study from the United Kingdom found that employment of either parent was associated with improved child nutritional outcomes, compared to both parents being unemployed [21]. Another study from China found that associations differed depending on which parent was employed: paternal unemployment was negatively associated with child health, while maternal unemployment was positively associated with child nutrition and health [22]. However, this nascent literature from upper-middle and high-income countries may not be generalizable to LLMICs, where employment opportunities are more informal and of lower wages [23] and where agricultural employment is the predominant type of labour (49% in LLMICs vs. 21% in upper-middle income countries vs. 3% in high-income countries [5]).

These mixed findings are not surprising given the complexity of the relationships between parental occupation and child nutritional outcomes, varying study methodologies, and the competing mechanisms through which parental employment and occupation type can influence child outcomes (see Section 1.2). Given that family and ecological caregiving environments are similar for child nutrition and development [24], associations between parental occupation and child development are plausible and equally important to unpack.

### 1.2. Hypothesised mechanisms

Parental occupation can influence child development through five potential mechanisms: (1) household income, (2) women's empowerment, (3) childcare, (4) parental physical and mental health, and (5) pesticide exposure. The relative importance of these mechanisms likely varies

by occupation type. In LLMICs, agricultural employment is common with 60% of working adults (63% of women, 57% of men) in low-income countries employed in agriculture and 38% (42% of women, 36% of men) in lower-middle income countries [5].

First, parental employment increases household income [11, 25], and thus the availability of additional resources to allocate towards child health, nutrition, and development [26, 27]. Families with higher income may have more and better access to child healthcare services in times of child illness [28] and financially provide more enriching opportunities for early learning like preschool fees and toys [1].

Second, parental employment can improve child development by increasing women's empowerment. Extensive empirical literature from LLMICs has demonstrated that women's employment is associated with greater women's empowerment [29–31]. In fact, women's employment, seasonality of employment, and type of remuneration (cash vs. in-kind) are often included as indicators in composite measures of women's empowerment [27, 32, 33]. Further, evidence suggests that women's non-agricultural employment is associated with greater women's empowerment relative to agricultural employment [34–36], likely because non-agricultural employment allows women to learn non-farm skills and exposes them to knowledge and information which can improve their household decision-making [34]. More empowered women allocate more resources towards their children, which in turn support better child development [27, 37]. Of note, prior literature on women's empowerment has largely been in the context of women's employment, despite scholars highlighting the need for research on employment and empowerment to include both men and women [38].

Third, parental employment can influence child development by impacting childcare arrangements, such as reducing the amount of time parents spend with their children and/or increasing reliance on alternative caregivers within (e.g., other adults or children) or outside (e.g., preschool) the household [7, 11, 39]. Given the morning and seasonal nature of farming, parents may have limited time with their children and require alternative non-parental childcare (e.g., supervision by older siblings). Evidence indicates different time patterns in childcare between working and non-working parents [6, 40–43]. However, the relationship between parental childcare time patterns and child development is complex and can vary depending on how parents manage time trade-offs and negotiate caregiving activities [43]. Studies of women's time use in agriculture show that women in agricultural settings face more severe time constraints and stricter trade-offs than women in non-agricultural settings [44, 45], which could adversely affect child development. Finally, considering that fathers are the primary breadwinners in many LLMICs contexts and can positively influence child development [46], there may also be interactions between maternal and paternal occupation with respect to childcare. However, evidence considering both maternal and paternal work and childcare is lacking from LLMICs.

Fourth, employed women have to balance paid work and unpaid childcare and household work, which can compromise their physical and mental health and, in turn, affect their ability to care for themselves and their children [40]. The potential adverse effects of employment on parental health are likely not unique to women, though they are probably exacerbated relative to men given employed women's dual burden of occupational and household work [47]. Stark gender inequities in the distribution of other household responsibilities remain, with women largely responsibility for all household chores, including physically intensive chores like collecting water and firewood [40]. Moreover, parental health is likely worse among parents employed in agriculture, which is generally more physically taxing than other occupations.

Lastly, pesticide exposure is a unique mechanism through which agricultural work can influence child development. Pesticides are widely used in agricultural contexts in LLMICs, with higher exposure due to continued use of harmful pesticides banned in high-income

countries and unsafe handling and application practices [48]. Children of parents engaged in agriculture can be exposed to pesticides by: (1) spending time on the farm where they may inhale pesticides directly during spraying or ingest them by touching objects and putting them or their fingers in their mouths; or (2) consuming contaminated foods or water [49, 50]. Extensive evidence has linked pesticide exposure to suboptimal child development, largely through inhibition of acetylcholinesterase activity [51, 52]. Pesticide exposure can also adversely affect parental health [53] and compromise parents' ability to care for their child.

### 1.3. Current study

Given the high proportion of developmentally off-track children in LLMICs and employed parents, unpacking the relationships between parental occupation and child development is crucial. Understanding the strength and direction of these relationships can help inform the design and targeting of interventions to support working parents with young children. Therefore, in this paper, we used nationally representative data from nine Demographic and Health Surveys (DHS) to investigate the associations between parental employment and child development, and the role of childcare and women's empowerment as potential mechanisms. Given the high proportion of adults employed in agriculture in LLMICs, we compare agricultural to non-agricultural employment. Based on our review of the literature and mechanisms (Sections 1.1 and 1.2), in this analysis, we tested five exploratory hypotheses. Due to the limited literature considering both maternal and paternal employment, we had no *a priori* hypotheses about parental sex and occupation type. Compared to non-agricultural employment, we hypothesised that parental agricultural employment is associated with:

1. poorer child development (with the poorest outcomes among families where both parents are employed in agriculture relative to both parents employed in non-agriculture)

2. lower women's empowerment

3. less child stimulation

4. less adequate supervision

5. less attendance in early childhood care and education programmes (ECCE)

We build on existing literature by examining child development as our primary outcome and considering both maternal and paternal occupation as determinants.

## 2. Methods

### 2.1. Data and study population

We pooled data from all DHS that collected information on parental employment and child development and were publicly available as of February 2022. All DHS collect data on women's employment in the last 12 months in the woman's interview. For a random sub-sample of households, an adult man is also interviewed, and the same employment questions are asked during the man's interview. A couples' file is then generated by the DHS Program pairing women and men who are married or cohabiting. With respect to child development, this optional module is applied to the youngest child aged 36–59 months, and collects data on children's attainment of developmental milestones, child stimulation, child supervision, and ECCE attendance. Since the child development module is optional, we could only include surveys which collected this module. However, within a DHS survey, the sub-samples of households completing the child development module and the man's interview are not always overlapping. Therefore, not all DHS surveys collecting data on child development contributed

data to our analysis. In total, we included nine DHS spanning 2011–2020: Benin, Burundi, Cambodia, Congo, Haiti, Rwanda, Senegal, Togo, and Uganda (**S1 Table**). These were all the countries with an overlapping sub-sample of households with data on child development and paternal employment.

## 2.2. Measures

We created two variables for parental employment and occupation in the last 12 months. First, we created a binary variable for whether each parent was employed in the last 12 months (employed vs. unemployed). Second, for those employed in the last 12 months, we created a categorical variable for whether one or both parents were employed in agriculture (occupation): (1) mother employed in agriculture, father employed in non-agriculture, (2) mother employed in non-agriculture, father employed in agriculture, (3) both parents employed in agriculture, and (4) both parents employed in non-agriculture. Non-agricultural occupations included: clerical, sales, services, professional/technical/managerial, household and domestic, skilled manual, and unskilled manual occupations. These groupings are pre-specified by the DHS Program [54].

Child development was assessed using the Early Childhood Development Index (ECDI), a population-based measure designed to assess four domains of development in children aged 36–59 months: cognitive, socio-emotional, literacy-numeracy, and physical [55]. The child's mother reports on whether the child can perform each of ten developmental milestones. Per the ECDI guidelines, we first created binary indicators for whether children attained each one of the ten developmental milestones. We then constructed binary indicators for whether children were developmentally on-track in each domain and for whether children were overall developmentally on-track (on-track in at least three out of the four domains). We also constructed the count ECDI score as the number of milestones achieved by each child, range 0–10 [55].

Child stimulation was assessed using three indicators for the number of stimulation activities (range 0–6) provided by the mother, father, or other household members in the last three days (all based on maternal report) [56]. The six activities were: (1) reading books/looking at pictures, (2) telling stories, (3) naming/counting/drawing, (4) singing, (5) **taking** the child outside, and (6) playing with the child. The adequacy of child supervision over the last week was assessed using three indicators: (1) child was not left alone for >1 hour, (2) child was not left under the supervision of another child for >1 hour, and (3) child was not left alone or under the supervision of another child (referred to as "adequate supervision" for brevity) [56]. ECCE attendance was assessed using a single indicator for whether the child attended an organised learning or ECCE programme.

We assessed three dimensions of women's empowerment: (1) access to and control over **resources**, (2) decision-making, and (3) attitudes towards wife beating. Factor scores for each dimension were derived from a form-invariant model using confirmatory factor analysis. We also calculated a total women's empowerment score as the sum of the three dimensions' factor scores. The only indicator related to women's employment included in the derivation of the "access to and control over resources" dimension was an indicator for seasonality of employment (throughout the year vs. seasonal/**occasional**). Full details on the indicators comprising the domains and the derivation of the factor scores have been previously published [27].

## 2.3. Statistical analysis

We restricted the analytic sample to children aged 36–59 months with available data on child **development**. We merged the couples and child files to create a mother-father-child triad, and

we therefore refer to women and men as mothers and fathers. Since only 97 (1.11%) fathers were not employed in the last 12 months, we restricted the sample to households where the father was employed. We tested for differences between included and excluded households using a Wald test, considered significant at $p<0.05$. Excluded households with unemployed fathers were generally similar to included households with employed **fathers**, except for the former provided less stimulation (**S2 Table**). We further excluded households with missing data on parental occupation (N = 142, 1.64%). Women in households without data on parental occupation were less empowered and more likely to have no education compared to women in households with data on parental occupation (S2 Table). In addition, fathers in households without data on parental occupation were older and more likely to have no education than fathers in households with data on parental occupation. Excluded households were also more likely to live in an urban area and were somewhat wealthier than included households (S2 Table). The final analytic sample included 8,516 children aged 36–59 months with data on child development and maternal and paternal occupation.

We used generalised linear models to assess the associations of interest. First, given that all fathers in our sample were employed, we examined the associations between maternal employment and child development (the outcome) and maternal employment and child stimulation, child supervision, ECCE attendance, and women's empowerment (the potential mechanisms). Then, among employed parents, we examined the associations between parental occupation and child development, and parental occupation and child stimulation, child supervision, ECCE attendance, and women's empowerment. We used log-Poisson models for the binary variables (overall development on-track, child supervision, and ECCE attendance) and calculated unadjusted and adjusted relative risk (RR) and 95% confidence intervals (CIs). For child stimulation and women's empowerment, we used linear models and calculated unadjusted and adjusted mean differences (MD) and 95% CIs. Adjusted models controlled for child age and sex, maternal age and education, paternal age and education, household size, wealth, and location (urban vs. rural). Standard errors were clustered at the primary sampling unit level. Models were weighted for representativeness using country-specific weights. Missing data on mechanisms (N = 64; 0.75%) were imputed using country-specific mean imputation.

In models including the four-category variable for parental occupation, we considered the category "both parents employed in non-agriculture" as the reference category. We tested for equality across exposure categories using a Wald test. In all models, the Wald tests indicated differences across exposure categories (all $p$-values <0.001). Given this non-equivalence, we then re-estimated the models changing the reference category to "both parents employed in agriculture" to assess whether associations differed depending on which parent was employed in agriculture. Thus, for a given outcome, the models using the categorical exposure were the same, except for this change in the reference category to assess the different associations.

Given the role of child nutrition (safe and nutritious foods) in supporting child wellbeing [57] and evidence that severe childhood malnutrition can impair child development [58], we conducted a sensitivity analysis accounting for child wasting (defined as weight-for-height Z-score <-2 SD) in the analysis. We first examined biserial correlations between child wasting, parental employment, and child development in the pooled sample and separately by country. We then included child wasting as a control variable in the adjusted models for the associations between maternal employment, parental occupation, and child development. Of note, child wasting was not available for the sample of children with data on child development and maternal and paternal occupation in Cambodia and Congo. Therefore, these sensitivity analyses were conducted on a dataset pooling the remaining seven countries.

To explore whether adjusted associations between parental occupation and child development, child stimulation, child supervision, ECCE attendance, and women's empowerment

differed across household location (urban vs. rural), household wealth, maternal and paternal education, and country income level (lower income vs. lower-middle income), we included an interaction term between the categorical exposure variable and each of these modifiers. Given the exploratory nature of these analyses, interactions were considered significant at $p<0.10$ based on a Wald test. All analyses were conducted in Stata 17 [59].

### 2.4. Ethical considerations

Ethical clearance for each DHS is granted by the relative institutions in the respective country. DHS data are publicly available de-identified secondary data and thus exempt from further ethical review. Access and permission to use the data for the present analyses was granted by the DHS Program (http://www.dhsprogram.com).

## 3. Results

### 3.1. Sample descriptives

Children in our sample were 46.6 months of age on average and 49% were girls (**Table 1**). All fathers and 85% of mothers in our sample were employed in the last 12 months. Among employed parents, 44% of mothers and 53% of fathers were employed in agriculture. Child development was generally poor with 40% of children developmentally off-track. Childcare practices and women's empowerment were also suboptimal. Child, maternal, paternal, and household characteristics by maternal employment and parental occupation are shown in **S3 Table**. Both parents were engaged in agriculture in 35% of the poorest households and 3% of the wealthiest households. Both parents were engaged in non-agriculture in 8% of the poorest households and 38% of the wealthiest households (S3 Table). Overall, households where both parents were employed in agriculture appeared poorer and less educated than households where only one or neither parent was employed in agriculture. Further, both agricultural and non-agricultural occupations differed by parental education, household wealth, and household location (**S4 Table**). The proportion of mothers and fathers employed in agriculture was lower among more educated individuals, wealthier households, and urban households. Respectively, the proportion of parents employed in professional/technical/managerial, sales, or skilled manual occupations was higher among more educated individuals, wealthier households, and urban households (S4 Table).

Overall, 2.4% of children in our sample were wasted, ranging from 0.6% in Rwanda to 10% in Senegal. In the pooled sample, child wasting was significantly correlated with maternal employment (p<0.01); however, the magnitude of this correlation was small (-0.03). Child wasting was not correlated with child development (overall or by domain) in our sample (all p-values >0.3 and all correlations were between -0.01 and 0.01).

### 3.2. Associations between maternal employment and child outcomes

Maternal employment was not associated with child development in adjusted models (**Table 2**). Results were consistent in sensitivity analyses controlling for child wasting (RR 1.01 (95% CI 0.95, 1.08)). However, maternal employment was associated with more stimulation provided by fathers and other household members, but with less adequate child supervision (Table 2). Maternal employment was not associated with stimulation provided by the mother or ECCE attendance. Maternal employment was positively associated with women's total empowerment, access to and control over resources, and decision-making (Table 2).

**Table 1. Household, parental, and child characteristics of the 8,516 children aged 36–59 months in the analytic sample.**

| | Mean±SD or proportion |
|---|---|
| *Household characteristics* | |
| Size | 6.81±3.71 |
| Number of children <5 y | 2.03±1.1 |
| Lives in rural area | 75.73 |
| Is in poorest quintile | 24.24 |
| *Maternal characteristics* | |
| Age, years | 31.49±6.37 |
| Highest level of education | |
| None | 37.87 |
| Primary | 40.81 |
| Secondary or higher | 21.32 |
| Employment status | |
| Unemployed | 15.27 |
| Employed in agriculture | 44.07 |
| Employed in non-agriculture | 40.66 |
| *Paternal characteristics* | |
| Age, years | 37.16±7.84 |
| Highest level of education | |
| None | 28.00 |
| Primary | 43.54 |
| Secondary or higher | 28.46 |
| Employment status | |
| Unemployed | 0.00 |
| Employed in agriculture | 53.39 |
| Employed in non-agriculture | 46.61 |
| *Parental occupation* | |
| Mother employed in agriculture; father employed in non-agriculture | 11.96 |
| Mother employed in non-agriculture; father employed in agriculture | 15.71 |
| Both parents employed in agriculture | 40.06 |
| Both parents employed in non-agriculture | 32.27 |
| *Child characteristics* | |
| Male | 50.83 |
| Age, months | 46.61±7.11 |
| Overall development on-track | 59.98 |
| Early Childhood Development Index Score (range 0–10) | 5.3±1.76 |
| *Childcare practices* | |
| Number of stimulation activities provided by | |
| Mother | 1.77±1.77 |
| Father | 0.83±1.44 |
| Other household members | 1.69±1.96 |
| Supervision in the past week | |
| Child not left alone for >1 hour | 81.64 |
| Child not left under the supervision of another child for >1 hour | 71.2 |
| Child provided adequate supervision | 64.56 |
| Child attended an early childhood education programme | 24.39 |
| *Women's empowerment* | |

(*Continued*)

**Table 1.** (Continued)

|  | Mean±SD or proportion |
|---|---|
| Access to and control over resources | -0.04±0.92 |
| Decision-making | 0.38±0.69 |
| Attitudes towards wife-beating | -1.21±1.64 |
| Total empowerment | -0.88±2.21 |

## 3.3. Associations between parental agricultural employment and child development

Among employed parents, children whose parents were employed in agriculture relative to non-agriculture were less likely to be developmentally on-track (**Table 3**). Results were consistent in sensitivity analyses controlling for child wasting (RR 0.85 (95% CI 0.79, 0.92)). More specifically, overall ECDI score, cognitive, socio-emotional, and literacy-numeracy development were poorer among children whose parents were both employed in agriculture relative to non-agriculture (**S5 Table**). When assessing whether associations differed by which parent was employed in agriculture, we observed that children were more likely to be developmentally on-track if only one parent was employed in agriculture as compared to both, regardless of whether it was the mother or father (**S6 Table**). The one exception was that children were more likely to be cognitively on-track if only the father was employed in agriculture relative to both parents (S6 Table).

**Table 2. Associations between maternal employment and child development, childcare practices, and women's empowerment[1].**

|  | Employed vs. unemployed mothers | |
|---|---|---|
|  | **Unadjusted** | **Adjusted** |
| *Child development* |  |  |
| Overall development on-track† | 0.96 (0.91, 1.02) | 0.98 (0.92, 1.04) |
| *Childcare practices* |  |  |
| Number of stimulation activities provided by |  |  |
| Mother | -0.05 (-0.19, 0.10) | 0.04 (-0.11, 0.19) |
| Father | 0.07 (-0.04, 0.17) | 0.12 (0.02, 0.22) |
| Other household members | 0.16 (0.02, 0.31) | 0.16 (0.01, 0.30) |
| Supervision in the past week |  |  |
| Child not left alone for >1 hour† | 0.97 (0.93, 1.00) | 0.96 (0.92, 0.99) |
| Child not left under the supervision of another child for >1 hour† | 0.87 (0.83, 0.90) | 0.90 (0.86, 0.93) |
| Child provided adequate supervision† | 0.84 (0.80, 0.88) | 0.87 (0.83, 0.91) |
| Child attended an early childhood education programme† | 0.82 (0.73, 0.93) | 0.96 (0.86, 1.08) |
| *Women's empowerment* |  |  |
| Access to and control over resources | 1.75 (1.71, 1.79) | 1.77 (1.73, 1.81) |
| Decision-making | 0.33 (0.27, 0.40) | 0.26 (0.21, 0.32) |
| Attitudes towards wife beating | -0.18 (-0.31, -0.05) | -0.07 (-0.20, 0.06) |
| Total empowerment | 1.90 (1.74, 2.06) | 1.96 (1.80, 2.12) |

[1] All fathers in the analytic sample were employed. All estimates are mean difference (MD) and 95% CI unless specified otherwise. All models accounted for representativeness. SEs were clustered at the primary sapling unit level. Adjusted estimates controlled for child age and sex, maternal age and education, paternal age and education, household size, wealth, and location (urban vs. rural).
† Estimates are relative risks (RR) and 95% CI.

**Table 3. Associations between parental occupation and development among children aged 36–59 months[1].**

| | Overall development on-track | |
|---|---|---|
| | **Unadjusted RR** | **Adjusted RR** |
| Both parents employed in non-agriculture | Ref | Ref |
| Mother employed in agriculture; father employed in non-agriculture | 0.88 (0.81, 0.95) | 0.92 (0.85, 1.00) |
| Mother employed in non-agriculture; father employed in agriculture | 0.95 (0.89, 1.01) | 1.04 (0.97, 1.11) |
| Both parents employed in agriculture | 0.77 (0.73, 0.82) | 0.86 (0.80, 0.92) |

[1] All models accounted for representativeness. SEs were clustered at the primary sapling unit level. Adjusted estimates controlled for child age and sex, maternal age and education, paternal age and education, household size, wealth, and location (urban vs. rural). Abbreviations used: MD, mean difference; RR, relative risk

## 3.4. Associations between parental agricultural employment and hypothesised mechanisms

With respect to the potential mechanisms we examined, parental agricultural employment relative to non-agricultural employment was not associated with stimulation by mothers or fathers (**Table 4**). However, stimulation by other household members was higher (15% additional activities) when both parents were employed in agriculture compared to when both parents were employed in non-agriculture (Table 4). Stimulation by other household members was significantly lower when only one parent was employed in agriculture relative to both parents (**S7 Table**). In addition, stimulation by mothers was higher if only the mother was employed in agriculture relative to both parents (S7 Table). Further, parental agricultural employment relative to non-agricultural employment was associated with inadequate child supervision, particularly leaving the child with an older sibling (Table 4). Parental agricultural employment was also associated with less ECCE attendance than parental non-agricultural employment. These associations with child supervision and ECCE attendance were largely driven by paternal employment in agriculture (S7 Table).

In addition, we found that parental agricultural employment was negatively associated with women's empowerment relative to parental non-agricultural employment: in households where both parents were employed in agriculture, women had lower scores on total empowerment, access to and control over resources, and attitudes towards wife beating (**Table 4**). These associations appeared to be largely driven by maternal agricultural employment (**S9 Table**).

## 3.5. Heterogeneity of associations

We found that the magnitude of the associations between parental occupation (both in agriculture vs. both in non-agriculture) and child development was larger among less educated parents and in low income countries (p-values for interaction <0.10) (**Fig 1**, **S9 Table**). There was also evidence that household location (urban vs. rural), parental education, and country income level modified the associations between occupation and stimulation (p-values for interaction <0.10). We observed less beneficial associations for stimulation by the mother among rural households and more educated parents, a less beneficial association for stimulation by the father in lower-middle income countries, and a more beneficial association for stimulation by other household members among rural households and in low income countries (**Fig 2**, S9 Table). Further, household wealth and country income level modified the association between occupation and adequate supervision, whereas maternal education and country income level modified the association between occupation and ECCE attendance (p-

**Table 4. Associations between parental occupation and childcare practices and women's empowerment among children aged 36–59 months[1].**

*Panel A Childcare practices*

| | Number of stimulation activities provided by the mother | | Number of stimulation activities provided by the father | | Number of stimulation activities provided by other household members | | | |
|---|---|---|---|---|---|---|---|---|
| | Unadjusted MD | Adjusted MD | Unadjusted MD | Adjusted MD | Unadjusted MD | Adjusted MD | | |
| Both parents employed in non-agriculture | Ref | Ref | Ref | Ref | Ref | Ref | | |
| Mother employed in agriculture; father employed in non-agriculture | -0.16 (-0.33, 0.01) | 0.11 (-0.07, 0.286) | -0.34 (-0.47, -0.21) | 0.02 (-0.13, 0.16) | -0.05 (-0.24, 0.13) | 0.08 (-0.11, 0.27) | | |
| Mother employed in non-agriculture; father employed in agriculture | -0.45 (-0.63, -0.28) | -0.04 (-0.22, 0.14) | -0.34 (-0.47, -0.21) | 0.00 (-0.13, 0.14) | -0.13 (-0.3, 0.04) | -0.01 (-0.19, 0.17) | | |
| Both parents employed in agriculture | -0.50 (-0.64, -0.36) | -0.06 (-0.21, 0.09) | -0.40 (-0.51, -0.28) | -0.04 (-0.16, 0.09) | 0.11 (-0.04, 0.26) | 0.26 (0.09, 0.42) | | |
| | Child not left alone for >1 hour in the past week | | Child not left under the supervision of another child for >1 hour in the past week | | Child provided adequate stimulation | | Child attended an early childhood education programme | |
| | Unadjusted RR | Adjusted RR | Unadjusted RR | Adjusted RR | Unadjusted RR | Adjusted RR | Unadjusted RR | Adjusted RR |
| Both parents employed in non-agriculture | Ref | Ref | Ref | Ref | Ref | Ref | Ref | Ref |
| Mother employed in agriculture; father employed in non-agriculture | 1.02 (0.98, 1.07) | 1.01 (0.96, 1.05) | 0.81 (0.98, 0.86) | 0.84 (0.79, 0.9) | 0.84 (0.78, 0.90) | 0.86 (0.80, 0.93) | 0.49 (0.40, 0.58) | 0.66 (0.55, 0.79) |
| Mother employed in non-agriculture; father employed in agriculture | 0.95 (0.90, 0.99) | 0.96 (0.91, 1.00) | 0.86 (0.82, 0.91) | 0.91 (0.85, 0.96) | 0.85 (0.80, 0.91) | 0.90 (0.83, 0.96) | 0.62 (0.54, 0.72) | 0.95 (0.82, 1.09) |
| Both parents employed in agriculture | 1.01 (0.97, 1.05) | 1.01 (0.97, 1.05) | 0.76 (0.73, 0.8) | 0.81 (0.76, 0.85) | 0.79 (0.75, 0.83) | 0.83 (0.78, 0.88) | 0.27 (0.24, 0.32) | 0.46 (0.39, 0.54) |

*Panel B Women's empowerment*

| | Access to and control over resources | | Decision-making | | Attitudes towards wife beating | | Total empowerment | |
|---|---|---|---|---|---|---|---|---|
| | Unadjusted MD | Adjusted MD | Unadjusted MD | Adjusted MD | Unadjusted MD | Adjusted MD | Unadjusted MD | Adjusted MD |
| Both parents employed in non-agriculture | Ref | Ref | Ref | Ref | Ref | Ref | Ref | Ref |
| Mother employed in agriculture; father employed in non-agriculture | -0.48 (-0.55, -0.42) | -0.42 (-0.49, -0.35) | 0.03 (-0.04, 0.09) | -0.02 (-0.08, 0.04) | -0.96 (-1.12, -0.80) | -0.67 (-0.84, -0.51) | -1.41 (-1.60, -1.23) | -1.11 (-1.30, -0.91) |
| Mother employed in non-agriculture; father employed in agriculture | -0.05 (-0.13, 0.02) | 0.03 (-0.05, 0.10) | -0.04 (-0.10, 0.02) | 0.01 (-0.05, 0.06) | -0.22 (-0.37, -0.07) | 0.06 (-0.10, 0.23) | -0.31 (-0.52, -0.11) | 0.10 (-0.11, 0.31) |
| Both parents employed in agriculture | -0.48 (-0.54, -0.43) | -0.39 (-0.45, -0.33) | -0.04 (-0.09, 0.01) | -0.01 (-0.06, 0.04) | -0.95 (-1.06, -0.84) | -0.61 (-0.75, -0.47) | -1.47 (-1.62, -1.32) | -1.01 (-1.18, -0.84) |

[1] All models accounted for representativeness. SEs were clustered at the primary sapling unit level. Adjusted estimates controlled for child age and sex, maternal age and education, paternal age and education, household size, wealth, and location (urban vs. rural). Abbreviations used: MD, mean difference; RR, relative risk

values for interaction <0.10). However, no clear patterns appeared by household wealth quintile or maternal education level. With respect to country income level, associations generally appeared less beneficial among lower-middle income countries. Lastly, household location, household wealth, and country income level modified the associations between parental occupation and women's empowerment (p-value for interaction <0.10) with larger, more negative associations among rural household, wealthier households, and lower-middle income countries (Fig 2, S9 Table).

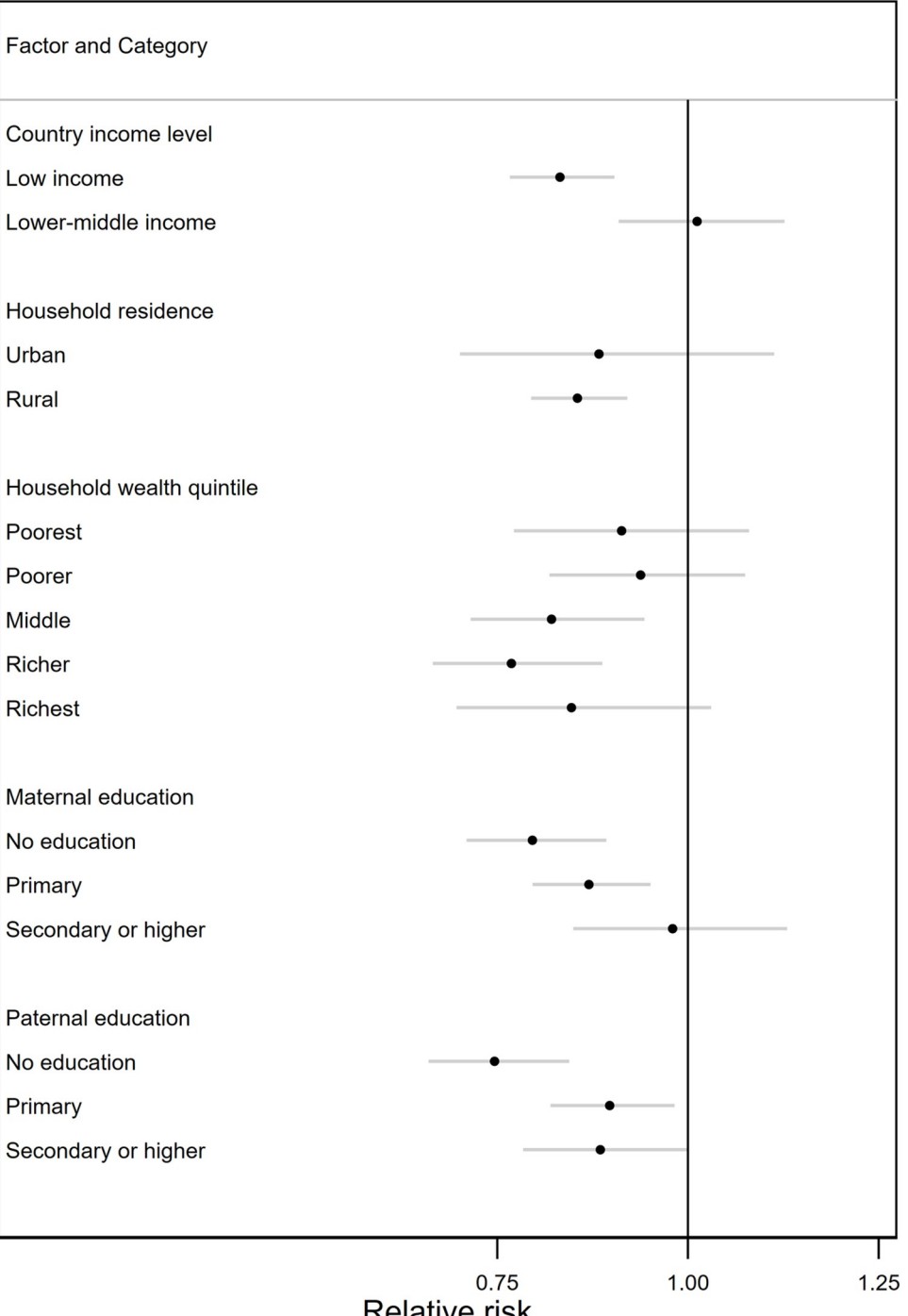

**Fig 1. Heterogeneity of the associations between parental occupation and child development by household and parental characteristics, comparing households where both parents were employed in agriculture and households where both parents were employed in non-agriculture.** Models adjusted for child age and sex, maternal age and education, paternal age and education, household size, wealth, and location (urban vs. rural), and accounted for representativeness. SEs were clustered at the primary sapling unit level.

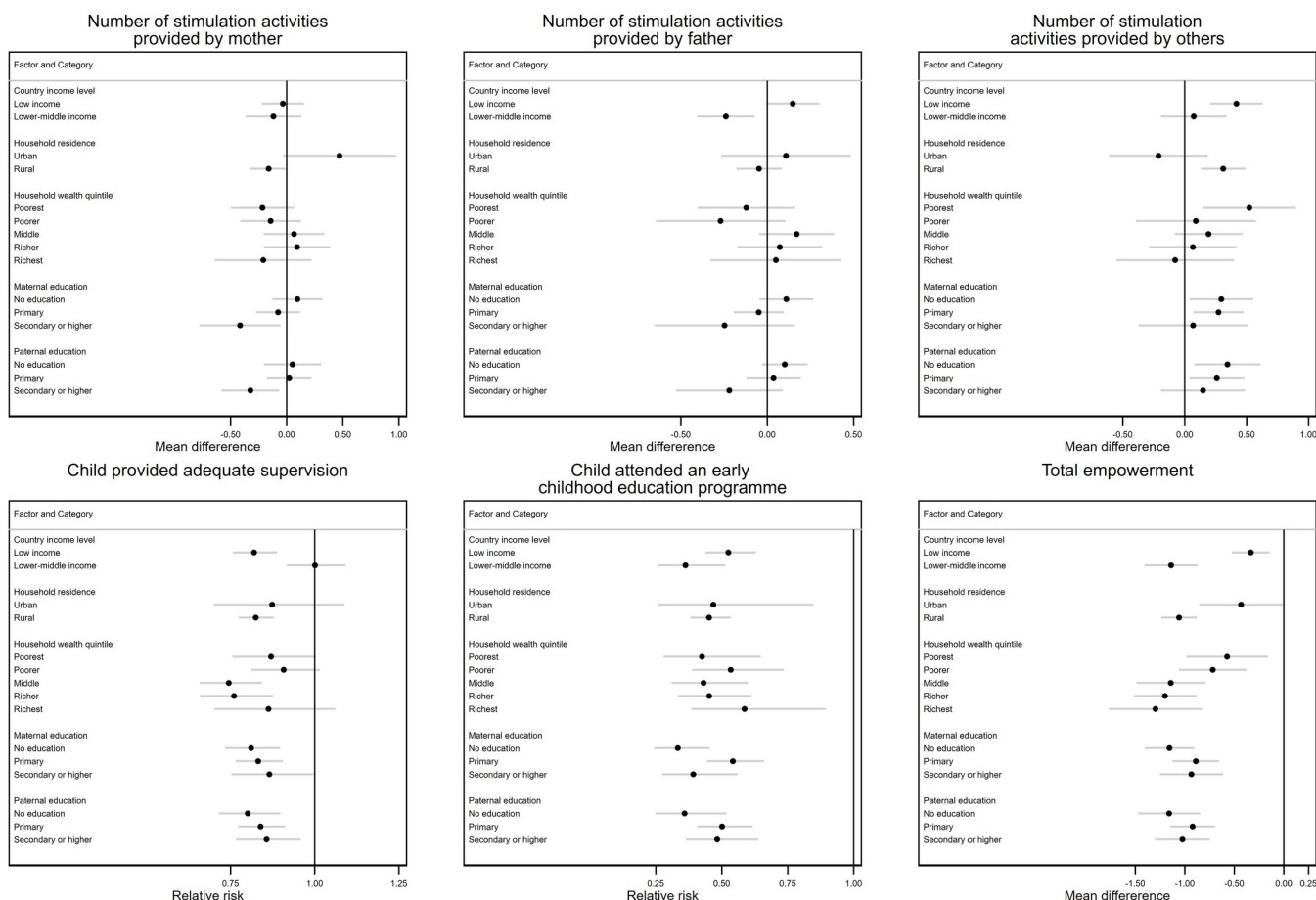

**Fig 2. Heterogeneity of the associations between parental occupation, childcare practices, and women's empowerment by household and parental characteristics, comparing households where both parents were employed in agriculture and households where both parents were employed in non-agriculture.** Models adjusted for child age and sex, maternal age and education, paternal age and education, household size, wealth, and location (urban vs. rural), and accounted for representativeness. SEs were clustered at the primary sapling unit level.

## 4. Discussion

In this study, we used nationally representative data from nine DHS surveys to investigate the relationship between parental employment (comparing agricultural and non-agricultural employment) and child development, and the role of childcare practices and women's empowerment as potential mechanisms. In support of four out of our five hypotheses, we found that parental agricultural employment, relative to non-agricultural employment, was associated with poorer child development, less adequate child supervision, less ECCE attendance, and lower women's empowerment. Contrary to our third hypothesis, we found that parental agricultural employment, relative to non-agricultural employment, was associated with more child stimulation provided by other household members. In exploratory analyses, we found that parental education, household wealth, and household location (urban vs. rural) modified the associations between parental occupation, child development, childcare practices, and women's empowerment.

Building on prior literature on the associations between maternal employment and child nutritional outcomes, we showed that any maternal employment was not associated with child development. However, given the mixed associations between maternal employment and

child nutritional outcomes [6–11, 15, 16], our results should be taken with caution until replicated using more comprehensive maternal employment measures (e.g., considering duration and hours worked) and better child development tools (e.g., based on direct assessment). With respect to the mechanisms we examined, consistent with existing literature we found that employed women had greater women's empowerment relative to unemployed women [29–31]. We further extend the evidence base by demonstrating that maternal employment was associated with certain caregiver practices, specifically with more child stimulation provided by fathers and other household members and less adequate supervision. These findings corroborate the hypothesis that employed women have to rely on alternative caregivers within the household [7, 11, 39]. It is worth noting that all fathers in our sample were employed. These relationships between maternal employment, child development, childcare practices, and women's empowerment may be different in families where fathers are unemployed, which may limit the generalisability of our findings to such families.

In a major extension of existing literature, our analysis demonstrated that parental agricultural employment was negatively associated with child development. Specifically, we found that children whose parents were employed in agriculture relative to non-agriculture had poorer overall, cognitive, socio-emotional, and literacy-numeracy development. As hypothesised, we observed the poorest child development outcomes when both parents were employed in agriculture vs. both employed in non-agriculture. Our analysis of potential mechanisms showed that both childcare practices and women's empowerment can help explain these differences in child development outcomes. First, with respect to caregiver practices, parental agricultural employment was associated with more stimulation provided by other household members and less adequate child supervision. These findings indicate that families where both parents are employed in agriculture are more reliant on alternative caregivers than families where both parents are employed in non-agriculture. However, given the cross-sectional nature of our analysis, it is also possible that other family members provide more childcare so that parents can farm, or that parents who farm delegate more childcare responsibilities to other family members. The presence and availability of potential alternative caregivers is likely to influence women's employment [39] and occupation type. Further, prior work on intergenerational transfer of childcare suggests that employed women transfer childcare to older female family members [40]. However, in our case, we were unable to determine which other family members provided more stimulation because of lack of data. It is plausible that older children provided more stimulation, a hypothesis substantiated by the increase in inadequate supervision we observed (specifically, leaving the child with an older sibling). However, given that families in our sample were relatively large (6.8 members on average), it is also possible that other family members such as mothers-in-law stepped in to provide more stimulation. Lastly, parental agricultural employment was associated with less ECCE attendance relative to non-agricultural employment. This difference may be due to differences in household wealth (with wealthier households better able to afford ECCE) or location (with better access in urban settings) [1]. However, our analysis of heterogeneity did not confirm this hypothesis.

Second, with respect to women's empowerment, we found lower women's empowerment scores in families where both parents were employed in agriculture relative to non-agriculture. Further, women employed in agriculture had lower women's empowerment scores than those employed in non-agriculture, results consistent with the literature [34–36], regardless of whether their partner was employed in agriculture or not. The limited role of paternal occupation in women's empowerment scores may be due to paternal employment status (employed vs. unemployed) being a more important determinant of women's empowerment than paternal occupation, or to the crude exposure indicator we used. The concurrent nature of the measures we used should also be noted. Women's empowerment is a process [60] and the

cumulative and/or lagged role of paternal occupation in shaping women's empowerment is theoretically and empirically unclear. More research is needed to better understand the role of men's employment in shaping women's empowerment trajectories both concurrently and over time [38].

We did not investigate household income, parental physical and mental health, and pesticide exposure as mechanisms in our analysis due to lack of data. First, DHS surveys do not collect data on household income. Instead, the DHS Program calculates a household wealth index representing fixed assets [54] that cannot easily be converted into resources for child development. Second, with respect to parental physical and mental health, DHS surveys collect data on maternal perinatal health with respect to the most recent pregnancy but do not collect data on current/general physical health, which is likely a more viable mechanism between parental employment and child development. A mental health module was introduced in DHS Phase 8 in 2020 and was thus not available for the surveys in our sample. Data on paternal physical and mental health is also lacking. Third, with respect to pesticide exposure, DHS surveys do not collect data on agricultural or occupational pesticide exposure. However, given the wide use of pesticides in LLMICs [48] and pesticides' direct adverse effects on child development through acetylcholinesterase inhibition [51, 52, 61], we cannot rule out that pesticide exposure was a major contributor towards the poorer child development outcomes we observed among agricultural families relative to non-agricultural families. Future work should collect data on these mechanisms to better understand their contribution towards child development and to help design interventions that support children and their families. Other early life biological and nutritional mechanisms beyond our proposed framework could also be explored. For example, evidence suggests that better diet in early infancy predicts improved development later in childhood [62, 63]. It is likely that families where both parents are engaged in agriculture rely more on staple foods, resulting in poor micronutrient and protein intake for themselves and their children. However, we lacked data on dietary intake in early childhood and were therefore unable to explore this mechanism. We considered the role of child wasting in sensitivity analyses and found no evidence that child wasting influenced the associations between maternal employment, parental occupation, and child development. This may be due to the low prevalence of child wasting in our sample (2.4%) or because undernutrition may be less harmful in children 36–59 months of age than in younger children [64]. Further, a growing body of evidence suggests that malnutrition in early childhood impairs cognitive, academic, and human capital outcomes later in childhood and adulthood [58, 65–67]. Due to the lack of data, we could only assess the role of concurrent child wasting. Although we found no evidence of a cross-sectional association between child wasting and development, future studies should consider previous and persistent episodes of child malnutrition, and if and how they may influence the associations between parental employment and child development.

Our analysis also uncovered some important potential modifiers of the associations between parental occupation and child development, childcare practices, and women's empowerment. Parental education modified the associations between parental occupation and child development, with somewhat more beneficial associations among educated parents. Educated parents may provide higher quality stimulation and early learning opportunities [68, 69], which could help explain the more beneficial child development outcomes in this subgroup. With respect to childcare practices, household location (rural vs. urban), household wealth, and parental education modified associations, but no clear patterns emerged. Further, these three factors also modified the associations between parental agricultural employment and women's empowerment. It appeared that women in wealthier households or with educated partners had lower empowerment scores than women in poorer households or with uneducated partners. However, occupation types for mothers and fathers differed across parental

education, household wealth, and household location categories (with non-agricultural occupations more common among more educated individuals, wealthier households, and urban households), which made these results difficult to interpret. Finally, country income level modified all associations we examined except for the association between parental agricultural employment and number of stimulation activities provided by the mother. However, patterns were inconsistent here as well with less beneficial associations for child development and adequate supervision in low income countries and for number of stimulation activities provided by the father, ECCE attendance, and women's empowerment in lower-middle income countries. Country income level likely reflects various country-level characteristics that influence these associations, such as the size of the agricultural sector, access to and quality of healthcare and early childhood care and education services, and gender equity. Importantly, these heterogeneity analyses were exploratory and hypothesis generating. Replication in samples adequately drawn and powered for subgroup analyses is needed before any definitive conclusions can be drawn. Future studies should examine other country-level characteristics that may moderate the associations we examined. Future work should also consider additional factors like parental use of alcohol, tobacco, and nicotine-containing products, which is prevalent in LMICs [70] and in farming populations [71], and associated with child development, particularly with poor emotional development and behavioural difficulties [72–74]. More evidence is needed on if, and how, parental substance use influences the relationship between parental occupation and child development in LLMICs. Due to data limitations on parental substance use and the fact that ECDI does not assess emotional development or behaviour, we did not include substance use in our analysis.

Among the strengths of our study was the use of nationally representative data from nine LLMICs and the large sample size. Nevertheless, several limitations are worth noting. First, the parental employment and occupation indicators we used were crude and captured limited information about parents' work. We had no data on employment duration (e.g., how many months or seasons parents were employed for), work schedule (e.g., day, night, weekend), numbers of hours work (e.g., full-time or part-time), or ability to bring the child to work. All these factors may influence childcare practices and women's empowerment, and in turn child development. For example, one study from Australia showed that parental nonstandard work schedules (e.g., evenings, nights, weekends) were associated with child overweight and obesity [75]. A study from Nigeria showed that children had worse nutritional outcomes when mothers did not bring them to work [11]. Studies using more comprehensive measures of parental employment encompassing all these aspects are needed to better understand the role parental employment plays in child outcomes. To help unpack additional mechanisms such as household income and parental physical and mental health, other occupation aspects should be considered including job security, regularity of payments, and working conditions.

Second, we only examined nuclear families, i.e., parents and their children. However, in many LLMICs, the concept of nuclear family extends to other household members and this is especially true for male caregivers [20, 76]. We lacked data on other family members and were unable to unpack how parental employment choices may influence or be influenced by the characteristics of other household members (e.g., age, caregiving roles, employment status). Relatedly, we largely examined mechanisms specific to mothers, i.e., maternal caregiving and empowerment. Child stimulation was the only mechanism (from the ones we examined) on which we had data for fathers, albeit the indicator was based on maternal report and thus subject to reporting bias. Given that both mothers and fathers can promote child development beyond stimulation practice alone [77], a wider consideration of other plausible mechanisms (e.g., sensitivity, positive disciplinary practices, emotional affect, perceived parenting stress)

obtained directly from both parents can improve our understanding of the relationship underlying parental employment and child development [78].

Finally, given the cross-sectional nature of the data, we were unable to establish causality or the temporal order of the exposure, mechanisms, and outcomes we examined. Thus, we were unable to conduct mediation analysis, formally test the mechanisms we examined, and quantity indirect effects. More longitudinal research is needed to establish the temporal order of the variables examined here and to assess if and how childcare practices and women's empowerment mediate the relationship between parental employment and child development. Relatedly, we could not resolve issues of endogeneity inherent in cross-sectional samples. Specifically, we could not determine the direction of the relationship between parental occupation type and parental education, household wealth, and household location. For example, we could not disentangle whether poorer parents chose to work in agriculture or whether those working in agriculture remained poorer. As a result, given that parental occupation type differed across parental education, household wealth, and household location, we were unable to conclusively demonstrate that the associations we observed were driven by parental occupation rather than by these other factors. We controlled for these factors in the adjusted models, which helped minimize confounding. However, we cannot rule out the presence of residual confounding. In addition, small sample sizes in sub-groups of households may have biased the results: both parents were engaged in agriculture in 3% of the wealthiest households, whereas both parents were engaged in non-agriculture in 8% of the poorest households. Replication in larger samples where more affluent and educated families are engaged in agriculture and less affluent and less educated families are engaged in non-agriculture is needed to provide more definitive support for our hypotheses.

Despite these limitations, we found suggestive evidence that maternal and paternal agricultural employment was associated with poorer child development, childcare practices, and women's empowerment. Our paper helps improve our understanding of the role of parental agricultural employment in shaping child development outcomes, childcare practices, and women's empowerment in LLMICs, thus filling important gaps in the literature. Nevertheless, to our knowledge, this is the first analysis of the relationship between parental agricultural employment and child development in LLMICs. Our analysis was largely exploratory and results should be taken with caution. Much research is still needed to fully unpack the complex relationships we examined and to help inform policies and interventions to support working parents with young children in LLMICs.

## Supporting information

**S1 Table. Demographic and Health Surveys (DHS) included in the sample.**
(DOCX)

**S2 Table. Differences between households included in the analytic sample and those excluded by exclusion reason.**
(DOCX)

**S3 Table. Child, maternal, paternal, and household characteristics by maternal employment and parental occupation.**
(DOCX)

**S4 Table. Maternal and paternal occupation type by parental education, household wealth, and household location.**
(DOCX)

**S5 Table. Associations between parental occupation and development domains among children aged 36–59 months.**
(DOCX)

**S6 Table. Associations between parental occupation and development among children aged 36–59 months.**
(DOCX)

**S7 Table. Associations between parental occupation and childcare practices among children aged 36–59 months.**
(DOCX)

**S8 Table. Associations between parental occupation and women's empowerment among children aged 36–59 months.**
(DOCX)

**S9 Table. Heterogeneity of the associations between parental occupation, child development, childcare practices, and women's empowerment by household location, household wealth, parental education, and country income level comparing both parents employed in agriculture vs. both parents employed in non-agriculture.**
(DOCX)

## Acknowledgments

We would like to thank all the participants in the studies. We are grateful to the DHS Program teams that implement, conduct, and complete the DHS surveys and make the data available.

## Author Contributions

**Conceptualization:** Lilia Bliznashka, Joshua Jeong, Lindsay M. Jaacks.

**Data curation:** Lilia Bliznashka.

**Formal analysis:** Lilia Bliznashka.

**Funding acquisition:** Lindsay M. Jaacks.

**Investigation:** Lilia Bliznashka.

**Methodology:** Lilia Bliznashka, Joshua Jeong, Lindsay M. Jaacks.

**Supervision:** Lilia Bliznashka.

**Validation:** Joshua Jeong, Lindsay M. Jaacks.

**Visualization:** Lilia Bliznashka.

**Writing – original draft:** Lilia Bliznashka, Joshua Jeong.

**Writing – review & editing:** Lilia Bliznashka, Joshua Jeong, Lindsay M. Jaacks.

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
