## [Decision Letter · Decision Letter 0]

4 Aug 2022

PGPH-D-22-01028

Maternal and paternal employment in agriculture and early childhood development: a cross-sectional analysis of Demographic and Health Survey data

Dear Dr. Bliznashka,

Thank you for submitting your manuscript to PLOS Global Public Health. After careful consideration, we feel that it has merit but does not fully meet PLOS Global Public Health’s publication criteria as it currently stands. Therefore, we invite you to submit a revised version of the manuscript that addresses the points raised during the review process.

Both reviewers point to serious gaps in the framing of the objectives and hypothesis, choice of analyses and the framing of the results. All of these need careful consideration among the study team and major revisions will be needed before this can be considered.

We look forward to receiving your revised manuscript.

Kind regards,

Prashanth Nuggehalli Srinivas, MBBS, MPH, PhD

Academic Editor

Journal Requirements:

1. We ask that a manuscript source file is provided at Revision. Please upload your manuscript file as a .doc, .docx, or .rtf.

2. Please provide separate figure files in .tif or .eps format and remove any figures embedded in your manuscript file. Please also ensure that all files are under our size limit of 10MB.

3. We notice that your supplementary tables are included in the manuscript file. Please remove them and upload them with the file type 'Supporting Information'. Please ensure that each Supporting Information file has a legend listed in the manuscript after the references list.

Additional Editor Comments (if provided):

Reviewers' comments:

Reviewer's Responses to Questions

**Comments to the Author**

1. Does this manuscript meet PLOS Global Public Health’s publication criteria? Is the manuscript technically sound, and do the data support the conclusions? The manuscript must describe methodologically and ethically rigorous research with conclusions that are appropriately drawn based on the data presented.

Reviewer #1: Partly

Reviewer #2: Partly

2. Has the statistical analysis been performed appropriately and rigorously?

Reviewer #1: Yes

Reviewer #2: Yes

3. Have the authors made all data underlying the findings in their manuscript fully available (please refer to the Data Availability Statement at the start of the manuscript PDF file)?

Reviewer #1: Yes

Reviewer #2: Yes

4. Is the manuscript presented in an intelligible fashion and written in standard English?

Reviewer #1: Yes

Reviewer #2: Yes

5. Review Comments to the Author

Reviewer #1: The study reports on maternal and paternal employment in agriculture and early childhood development from DHS data of 10 countries.The study concludes that parental agricultural employment was associated with poorer child development.The study is well conceptualised and written. However, before concluding that it is the occupation of agriculture, which a risk factor ,some issues need elucidation.

Major issues:1. All the countries reported(9 from Sub-Saharan Africa and Cambodia) have high levels of child stunting, wasting, anaemia and low birth weight.Since development is intimately linked to child growth and nutrition, it is likely that children from the agriculturists' families(which are poorer) are likely to have greater malnutrition.Child growth indicator data is available through DHS & can be incorporated in the analytic framework.

2. Dietary data, particularly diversity, adequacy and meal frequency can be extracted from DHS. It is likely that the agriculturists' families have a greater reliance on staple food cultivation and consumption with poor micronutrient and protein availability. This could help explain the study findings better.

Minor issues:

1.The authors do mention about pesticide exposure and its possible adverse effects.They understandably cannot report on it because of lack of data.However, tobacco use( smokeless as well as smoked) is widely prevalent in LMICs especially in the rural populations.A significant exposure to smokeless tobacco use is reported in even pregnant women in LMICs. This could adversely affect parental health and child development.

2. Alcohol use is also having significant prevalence among the rural poor in LMICs. This could be important in the context of wife beating, access and decision making, measured for women's empowerment, in the study.

3. Use of mosquito nets: many of these countries have endemicity of Malaria which is known to affect younger children far more than other age groups. Anemia as a consequence of Malaria could well pull down development scores. Use of mosquito nets is recorded in DHS and can be included in this analysis.

4. Lastly, since it is a multi-country study, there are likely to be significant differences between countries with respect to policies and provision of care like ECCE or health care facilities for women and children. This could be pertinent in the context of nutritional interventions, malaria, anaemia, tuberculosis or such tropical illnesses which have a wide impact on child development and growth.These differences can be mentioned in the study.

Reviewer #2: Summary

The objective of this primary research article was to determine the association between parental employment type and child development, and explore childcare practices and women's empowerment as potential mechanisms. 9 DHS surveys, primarily from Africa, were pooled for the analysis. Parental agricultural employment, as opposed to non-agricultural employment, was found to be a risk factor for child development outcomes, exposure to cognitive stimulation, attendance of early care and education services, inadequate supervision, and women's empowerment.

While the paper highlights the gaps in the literature and explains the rationale of the study very clearly, my main concern is related to the interpretation of the results. The authors have demonstrated that agricultural employment is more likely to be practiced by poorer, less educated individuals, who typically reside in rural areas. Therefore, one can argue that it is poverty, material deprivation, and the lack of adequate cognitive stimulation, rather than the type of employment, that is really driving these associations. Are similar results seen in populations where agricultural employment is practiced by more affluent and educated families?

Abstract:

The following statement is not valid - Our objective was to investigate the associations between maternal and paternal agricultural employment - since all types of employment were considered for analysis, not just agricultural employment. This statement is therefore biased towards which of the results showed significant differences.

Introduction:

The hypothesized mechanisms by which parental occupation type may affect child development outcomes are very well laid out. However, some of them were not tested (household income, parental physical and mental health). While the reason for not being able to examine pesticides exposure as a mechanism is indicated as a limitation later on, the reasons for leaving out the ones mentioned in the parenthesis are not clear. More so since data on both health and income are comprehensively collected in the DHS surveys.

The literature review is very well done. It flows well into the stated objectives.

Methods:

I am curious to know why the DHS surveys from these nine countries were specifically chosen for this study. Since this is a secondary data analysis project, why not include DHS data from all LLMICs surveys?

Given my concern above (poverty and material deprivation, rather than occupation type, is driving child development outcomes), can the authors conclusively demonstrate / prove their hypotheses? Could they consider a propensity score matching type of statistical method to resolve this?

Results:

Please clarify lines 290-293: Further, when both parents were employed in agriculture, they were less likely to provide adequate child supervision (particularly, not leave the child with an older sibling) compared to when both parents were employed in non agriculture (Table 4).

Discussion:

In Line 318, add 'surveys' after DHS

Line 319: Clarify that the authors intended to investigate the association between parental occupation type and child development...not "agricultural" employment alone

Hence, overall a very good read, but I am worried about the interpretation. I look forward to a rebuttal which will address my concerns.

6. PLOS authors have the option to publish the peer review history of their article (what does this mean?). If published, this will include your full peer review and any attached files.

**Do you want your identity to be public for this peer review?** For information about this choice, including consent withdrawal, please see our Privacy Policy.

Reviewer #1: **Yes: **Rama Krishna Sanjeev

Reviewer #2: **Yes: **Debarati Mukherjee

---

## [Decision Letter · Decision Letter 1]

17 Oct 2022

PGPH-D-22-01028R1

Maternal and paternal employment in agriculture and early childhood development: a cross-sectional analysis of Demographic and Health Survey data

Dear Dr. Bliznashka,

Thank you for submitting your manuscript to PLOS Global Public Health. After careful consideration, we feel that it has merit but does not fully meet PLOS Global Public Health’s publication criteria as it currently stands. Therefore, we invite you to submit a revised version of the manuscript that addresses the points raised during the review process.

We look forward to receiving your revised manuscript.

Kind regards,

Prashanth Nuggehalli Srinivas, MBBS, MPH, PhD

Academic Editor

Journal Requirements:

Additional Editor Comments (if provided):

Reviewers' comments:

Reviewer's Responses to Questions

**Comments to the Author**

1. If the authors have adequately addressed your comments raised in a previous round of review and you feel that this manuscript is now acceptable for publication, you may indicate that here to bypass the “Comments to the Author” section, enter your conflict of interest statement in the “Confidential to Editor” section, and submit your "Accept" recommendation.

Reviewer #1: All comments have been addressed

Reviewer #2: All comments have been addressed

2. Does this manuscript meet PLOS Global Public Health’s publication criteria? Is the manuscript technically sound, and do the data support the conclusions? The manuscript must describe methodologically and ethically rigorous research with conclusions that are appropriately drawn based on the data presented.

Reviewer #1: Partly

Reviewer #2: Partly

3. Has the statistical analysis been performed appropriately and rigorously?

Reviewer #1: No

Reviewer #2: Yes

4. Have the authors made all data underlying the findings in their manuscript fully available (please refer to the Data Availability Statement at the start of the manuscript PDF file)?

Reviewer #1: Yes

Reviewer #2: Yes

5. Is the manuscript presented in an intelligible fashion and written in standard English?

Reviewer #1: Yes

Reviewer #2: Yes

6. Review Comments to the Author

Reviewer #1: The Leroy et al paper referred to is on the relationship of stunting and child development & not on wasting.1 There is substantial published evidence indicating association between wasting (or acute child malnutrition) and even stunting with child development.2,3 While the authors’ state that there is no evidence that child growth and development are causally related, there is a significant association between the two.

Moreover, it is entirely plausible that a growth retarded and wasted child takes longer time to care & feed. There is also a likelihood of developmental delay in such a situation. Hence, the set of variables are complex and interrelated. Also, as per the well-established UNICEF framework for malnutrition, both nutrition and care are contributory to child growth and well-being.4

Studying the association between parental employment and child development cannot ignore this other major contributor, namely, malnutrition, as measured through anthropometry. The authors have also included nutrition as one of the agencies through which parental employment changes child development(under household income).

This is all the more significant in view of the authors’ conclusions relating agricultural work to child development. It should be noted that much of the food in Africa and Asia is produced by small holder farmers who ironically are the worst affected by malnutrition.5,6

It will be appropriate to use malnutrition related variables ,particularly wasting prevalence, as a contributory factor to child development in the analysis.

Even while considering both as outcomes of similar variables, it will be preferable to check for multicollinearity of anthropometric variables of various groups of families with respective child development, in the individual(disaggregated) country settings.

Line 344-349 sentence is too unwieldy and long.

References:

1. Leroy JL, Frongillo EA. Perspective: What Does Stunting Really Mean? A Critical Review of the Evidence. Adv Nutr. 2019 Mar 1;10(2):196-204. doi: 10.1093/advances/nmy101. PMID: 30801614; PMCID: PMC6416038.

2. Kirolos A, Goyheneix M, Kalmus Eliasz M, et al. Neurodevelopmental, cognitive, behavioural and mental health impairments following childhood malnutrition: a systematic review. BMJ Glob Health. 2022;7(7):e009330. doi:10.1136/bmjgh-2022-009330

3. Alam MA, Richard SA, Fahim SM, et al. Impact of early-onset persistent stunting on cognitive development at 5 years of age: results from a multi-country cohort study. PLoS One 2020;15:e0227839.

4. Black RE, Allen LH, Bhutta ZA, et al. Maternal and child undernutrition: global and regional exposures and health consequences. Lancet. 2008;371(9608):243-260. doi:10.1016/S0140-6736(07)61690-0

5. FAO. The state of food and agriculture: Innovation in family farming. Rome: Food and Agriculture Organization of the United Nations; 2014.

6. Sibhatu KT, Qaim M. Rural food security, subsistence agriculture, and seasonality. PLoS One. 2017 Oct 19;12(10):e0186406. doi: 10.1371/journal.pone.0186406. PMID: 29049329; PMCID: PMC5648179.

Reviewer #2: Thank you for addressing the reviewers comments. While I am satisfied with most of the responses, I would like to request for more clarity / additional analyses on the following point:

1. Regarding the author's point: First, DHS surveys do not collect data on household income. Instead, the DHS Program calculates a household wealth index representing fixed assets that cannot easily be converted into resources for child development.

- While I agree that household assets are not a good reflection of resources for child development, one could argue that they are a proxy measure, since those with higher number of household assets may also be able to afford more resources related to children's development.

Please can I request the authors to discuss why they may be seeing a bigger difference between agricultural vs non-agricultural employment in the richer/richest households compared to the poorest households with respect to "on-track" development. It would also help to highlight the sample size in each of these categories. Eg: how many families have both parents employed in the agricultural sector in the richest vs poorest categories. I just want to make sure that small numbers are not leading to misinterpretation of the results.

Satisfied with all other responses - and the more cautious interpretation of the results in the discussion.

7. PLOS authors have the option to publish the peer review history of their article (what does this mean?). If published, this will include your full peer review and any attached files.

**Do you want your identity to be public for this peer review?** For information about this choice, including consent withdrawal, please see our Privacy Policy.

Reviewer #1: **Yes: **Dr Rama Krishna Sanjeev

Reviewer #2: **Yes: **Debarati Mukherjee

---

## [Decision Letter · Decision Letter 2]

12 Dec 2022

Maternal and paternal employment in agriculture and early childhood development: a cross-sectional analysis of Demographic and Health Survey data

PGPH-D-22-01028R2

Dear Dr Bliznashka,

We are pleased to inform you that your manuscript 'Maternal and paternal employment in agriculture and early childhood development: a cross-sectional analysis of Demographic and Health Survey data' has been provisionally accepted for publication in PLOS Global Public Health.

Best regards,

Prashanth Nuggehalli Srinivas, MBBS, MPH, PhD

Academic Editor

Reviewer Comments (if any, and for reference):

Reviewer's Responses to Questions

**Comments to the Author**

1. If the authors have adequately addressed your comments raised in a previous round of review and you feel that this manuscript is now acceptable for publication, you may indicate that here to bypass the “Comments to the Author” section, enter your conflict of interest statement in the “Confidential to Editor” section, and submit your "Accept" recommendation.

Reviewer #1: All comments have been addressed

Reviewer #2: All comments have been addressed

2. Does this manuscript meet PLOS Global Public Health’s publication criteria? Is the manuscript technically sound, and do the data support the conclusions? The manuscript must describe methodologically and ethically rigorous research with conclusions that are appropriately drawn based on the data presented.

Reviewer #1: Yes

Reviewer #2: Yes

3. Has the statistical analysis been performed appropriately and rigorously?

Reviewer #1: Yes

Reviewer #2: Yes

4. Have the authors made all data underlying the findings in their manuscript fully available (please refer to the Data Availability Statement at the start of the manuscript PDF file)?

Reviewer #1: Yes

Reviewer #2: Yes

5. Is the manuscript presented in an intelligible fashion and written in standard English?

Reviewer #1: Yes

Reviewer #2: Yes

6. Review Comments to the Author

Reviewer #1: Comments have been addressed adequately.

Reviewer #2: All my comments have now been addressed.

7. PLOS authors have the option to publish the peer review history of their article (what does this mean?). If published, this will include your full peer review and any attached files.

**Do you want your identity to be public for this peer review?** For information about this choice, including consent withdrawal, please see our Privacy Policy.

Reviewer #1: **Yes: **Rama Krishna Sanjeev

Reviewer #2: **Yes: **Debarati Mukherjee
